# Learning Activities with Plants and Technology: A Systematic Literature Review

**Alejandro Leo-Ramírez** [1,*], **Jose Alvarez** [2], **Marina Pérez** [3], **Wolfgang Greller** [4] **and Bernardo Tabuenca** [1]

1   Departamento de Sistemas Informáticos, ETSISI, Universidad Politécnica de Madrid, 28031 Madrid, Spain
2   Departamento de Ingeniería Agroforestal, ETSIAAB, Universidad Politécnica de Madrid, 28040 Madrid, Spain
3   Escuela Politécnica, Universidad Alfonso X El Sabio, 28691 Madrid, Spain
4   Institute for Interdisciplinary Teacher Education Vienna, University College of Teacher Education, 1100 Vienna, Austria
*   Correspondence: alejandro.leo.ramirez@upm.es

**Abstract:** The increase in remote work and study modalities in recent years has changed our indoor physical spaces. Key variables such as air quality, temperature, or well-being in general have acquired special relevance when designing workspaces. In this context, plants can play an active role in moderating these variables and providing well-being to the people who live in these spaces. UNESCO, when framing its 2030 agenda, highlighted the importance of promoting environmental awareness at the educational level (Sustainable Development Goals 3, 4, and 11). The scientific literature shows that the potential of plants is not sufficiently well addressed in educational contexts. Therefore, this review explores activities in which plants are used as a deliberate object of attention in learning contexts. The results show what learning activities have been carried out, what kind of plants have been used in the activities, and what technologies have supported those activities. The results provide a clear vision of the potential of plants to naturalize indoor learning spaces and to promote environmental awareness. This work aims to provide cues for further research on green education towards a sustainable society.

**Keywords:** learning activities; plants; technology-enhanced learning; well-being





## 1. Introduction

Apart from the technological challenges connected to lockdowns and remote working, the COVID19 pandemic has brought with it also the importance of caring for nature and the importance of environmental awareness. This coincides with what is encompassed in the United Nations' Sustainable Development Goals (SDGs) [1], where several of the sections refer to the importance of sustainability and environmental awareness. However, the current use of technology in education is quite disconnected from the goal of environmental awareness and especially the protection of flora and fauna. The possible lack of a relationship between technology and plants means that the blending of both disciplines is sometimes difficult to find and has been severely underexplored.

Plants can play two different roles in learning activities: (a) a primary role, where plants are used during the learning activity (foreground); or (b) a secondary role, where plants do not intervene in the learning activity (background). This review classifies learning activities where plants had a primary role. Therefore, studies investigating the effects of plants in the classroom where plants were not a deliberate object of attention during the learning activity, but rather the subject of an independently performed research, were discarded (e.g., the impact of plants on psychological [2] or environmental aspects [3]).

On the technological side, researchers have tried to bring together new trends in the use of technologies in education, as can be seen in the 2022 EDUCAUSE Horizon Report by Pelletier et al. [4]. Thus, education is relying on emergent technologies to provide a better

classroom experience [5,6]; among others, Internet of Things (IoT), learning analytics, artificial intelligence, virtual reality, augmented reality, and blockchain are trending educational technologies [4].

One of the emerging technologies that favors the creation of smart learning environments is IoT, which enable learning environments and learners to take advantage of features such as hypersituation [7] or student tracking [8]. Additionally, microcontroller and sensor programming learning activities are becoming more frequent [9–11] (thus promoting IoT), and the number of articles, conferences, and journals published on this topic has increased in recent years. However, education is not only taking advantage of IoT and its features to move towards smart learning spaces. There are meta-reviews that examine how other types of technologies have been used in education: big data [12], virtual reality [13], learning analytics [14], artificial intelligence [15], and blockchain [16].

While research into emergent learning technologies is thriving, the use of technology for learning about the natural environment or plants and their properties is not sufficiently explored in the scientific literature. The studies and reviews mentioned above do not focus on the field of botany and environmental awareness, but give an overview applied to all areas of education, bringing together all disciplines covered by education. This could represent a gap in the current research, given that there are technologies that can be applied specifically to a single subject such as biology and which better explore its characteristics. Therefore, we find it necessary to review what technologies have been used in teaching about plants, so that this can be a starting point to develop new learning activities where technology and plants can contribute jointly to teaching and learning.

The rest of this article is organized as follows. Section 2 describes the methodology applied to perform the literature review. Next, Section 3 describes the results of the analysis of relevant publications classifying learning activities, plants, and technologies and gaps to be addressed in further research are discussed. Finally, Section 4 concludes this article.

## 2. Methods

The method specified by Kitchenham & Charters [17] was followed to perform this SLR to ensure paper quality and reduce research biases. This method was conceived for the field of software engineering. However, its use has spread to multiple research areas, including Technology-Enhanced Learning [18,19].

### 2.1. Research Questions

This work reviews learning activities using plants and technology. With regard to the pedagogical approach, we have classified learning activities considering the type of activity, educational level, physical space, purpose, role of stakeholders, topic/subject area, and the variables observed. With regard to the technology used, articles are classified considering their characteristics (i.e., hardware/software, tools involved). Finally, plants are classified considering their species and the characteristics observed during the learning activity. Hence, the research questions (RQ) were formulated as follows:

1. RQ 1: What learning activities involving plants and technology have been carried out?
2. RQ 2: What technologies were involved in those learning activities?
3. RQ 3: What plant species were observed in those learning activities?

### 2.2. Selection of Studies

Five scientific bibliographic sources were selected to scan the scientific literature for this review: ACM, IEEEXplore, Scopus, ScienceDirect, and Web of Science (WoS). These databases cover the investigated disciplines: education, computer engineering/science, and plants. The selected date range was from January 2000 to October 2021 (the month this review started). Book chapters, conference papers, and research articles were considered in this review. The search string was: title ('plant' OR 'classroom') AND abstract ('plant' AND 'classroom') AND full text ('plant' AND 'classroom'). Figure 1 presents a diagram explaining the process of selection and filtering of scientific articles carried out in this review.

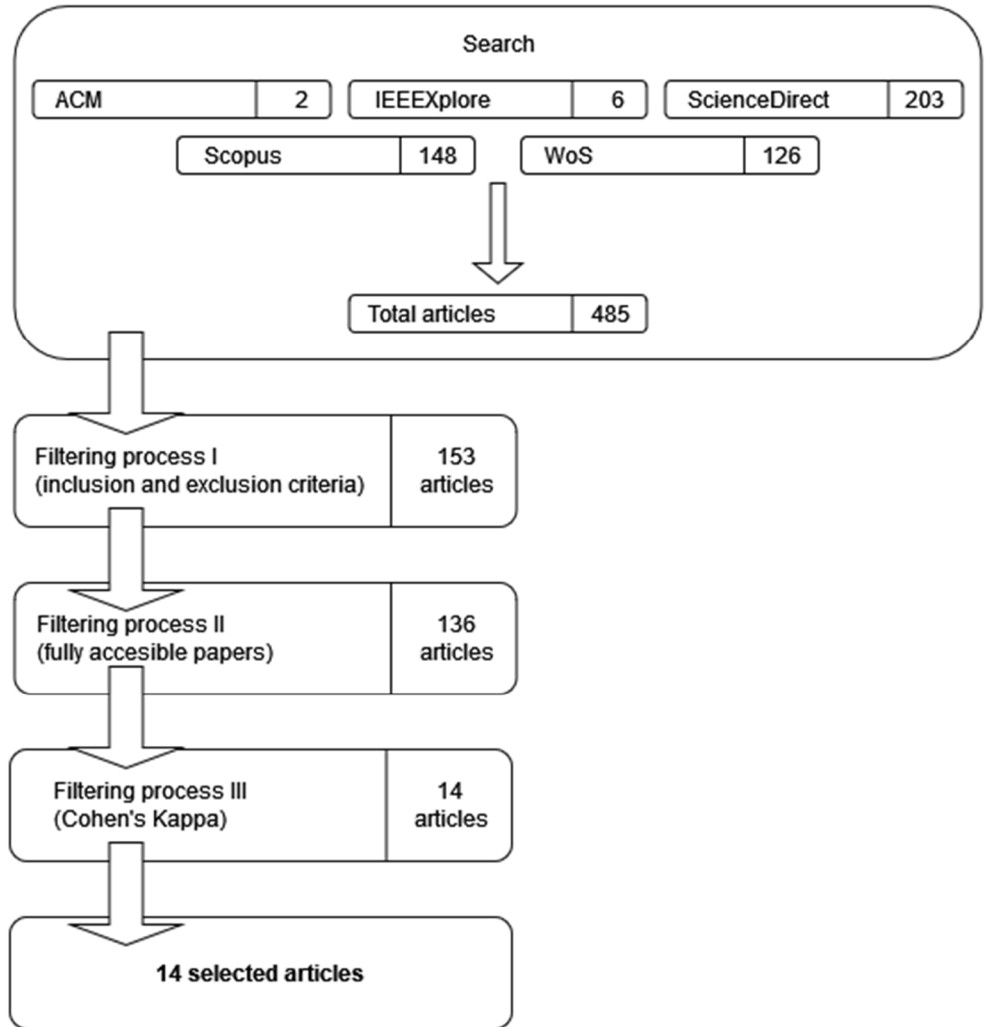

**Figure 1.** Methodology used in this systematic literature review.

Article filtering was performed to ensure the quality of the review as illustrated in Figure 1. The initial search considered the above-described query string, resulting in a total of 485 papers. Afterwards, the inclusion and exclusion criteria described in Table 1 were defined for filtering process I. During this process, the title, abstract, conclusion, and figures of the paper were analyzed. Duplicate papers were eliminated. This process resulted in a total of 153 papers.

Non-accessible articles (17 out of 153) were discarded after sending a request to the authors via Academia.edu and ResearchGate. This process (Filtering process II) resulted in a total of 136 papers.

The article selection (Filtering process III) process was optimized following the guidelines set by Pérez et al. [20]. In order to minimize bias during the selection process, Pérez et al. [20] suggest the following protocol based on the Cohen's Kappa statistic. Initially, 15 papers are chosen at random to perform a double review. Through an iterative process, the inclusion and exclusion criteria are refined considering the matches between the reviewers. A discussion is organized after every iteration to reach a consensus and refining the IC/EC criteria. The almost perfect agreement is achieved when k > 0.8, ensuring that each reviewer understands the review criteria in the same way. After achieving this agreement, the double review is eliminated, and each reviewer carries out the selection process independently. In this review, exclusion criteria E6-E9 had to be added after the first iteration (k = 0.65). Two iterations were necessary to reach agreement (k = 0.84). A total of 14 articles were finally selected.

**Table 1.** Inclusion and exclusion criteria.

| Inclusion Criteria | Exclusion Criteria |
|---|---|
| [I1] Empirical work. Tools that are evaluated in case studies. Papers describing systems/prototypes that are evaluated in lab or real contexts. | [E1] Off-topic papers. Publications were excluded if their main focus was not the use of technology for learning/teaching using (vegetable) plants. |
| | [E2] Publications focused exclusively on theories, philosophical aspects, concepts, visions, or ideas. Surveys on these aspects are not considered empirical papers. Non-empirical work. |
| | [E3] In the case of multiple articles reporting the same study, all but the most recent one were discarded. |
| | [E4] Publications not written in English. |
| | [E5] Articles with unclear contributions, poorly written articles that were difficult to understand, and articles with insufficiently described proposals. |
| | [E6] Papers that do not describe a learning activity using plants and technology. |
| | [E7] Technology (hardware, software or digital systems) is not explicitly mentioned in the text. |
| | [E8] Technology is not used by the subjects (stakeholders) of the experiment in the learning activity. |
| | [E9] The use of technology as an excuse to get the article published. |

### 2.3. Data Extraction

The data extraction process and analysis of the data was performed considering the three core themes as described in the research questions: pedagogical approach, technology, and plants.

Within each of these main categories, more specific subcategories were established from which to extract information. Consequently, a main category such as 'learning activity' was divided into subcategories such as 'Pedagogical approach', 'Educational level' or 'Purpose of the learning activity', among others. Supplementary Materials contains all the tables (Tables S1–S8) with the original data extracted.

In the data extraction process, the authors tried to be as precise as possible, which resulted in many rows of data in each of the tables. Therefore, in favor of a simpler reading and interpretation of the data, it was decided to reduce the number of rows by making these categorizations more general. A specific case of this could be to merge the categories 'Learning about plants, soil, water, and agricultural concepts', 'Learn about invasive species and plant biosecurity', 'Learn about pollen' and 'To explore the effects of biochar on plant growth and soil respiration' into a single category: 'Learning about plants, soil, water, and agricultural concepts'. In this way, different definitions that could come from the same root were brought together and the number of categories was reduced.

## 3. Results and Discussion

### 3.1. Analysis of the Selected Articles

Looking at the type of publication, 92.86% were journal articles [21–32] and 7.14% were conference/proceedings papers [33]. The articles were published evenly between 2007 and 2020, without identifying a particular trend. The most cited articles are [23,26,32]

with 346, 160, and 134 citations (as of February 2023), respectively. No prominent authors on this topic have been identified.

### 3.2. Learning Activities Involving Plants and Technology (RQ1)

This section aims at characterizing the learning activities performed using plants and technology. Here, learning activities are classified considering the following characteristics: education level, pedagogical approach, topic/subject of the learning activity, purpose of the learning activity, and educational space.

### 3.2.1. Education Level

Learning activities were classified considering the education level at which they were developed. The results show that 'Secondary school' [21,23,25,27,28,30,31,33,34] was the educational level at which most learning activities were conducted (64.29%), followed by 'Primary school' (50%) [22,24,26,29,30,32,34], and 'Tertiary school' [28]. Some learning activities were conducted [28,30,34] with students from two different educational levels.

With regard to secondary school levels, Southgate, E. et al. [21] conducted a virtual reality group activity (in groups of 3) in which 54 secondary school students participated. In the activity, they used immersive augmented reality with the Oculus Rift device in the videogame Minecraft to build a model of a plant to demonstrate their knowledge and understanding of different concepts related to plants such as respiration and photosynthesis. For this learning activity, they used a virtual species of *Hyacinthoides* (genre).

With regard to primary school levels, Liu, W. et al. [32] prepared an individual mixed reality (MR) learning activity in which 40 primary school students used the tool Plant Mixed Reality System (PMRS) to interact in a virtual environment with objects that could be picked up and moved. Students were able to perform different actions and explore key processes in a configurable virtual environment using a virtual plant: seed germination, seed disposal, photosynthesis, and reproduction. The virtual environment was touchless (without keyboard or mouse). The PMRS was intended to understand students' acceptance of the mixed reality (MR) technology for learning, and the factors pertinent to influence their intention to use it. In addition, a quiz was included in the system to measure "entertainment" factors and to evaluate the system. The PMRS was considered interactive, enjoyable, interesting, and engaging.

With regard to mixed school levels, Silva, H. et al. [28] carried out a group activity (2 students per group) in which a total of 296 secondary school and tertiary school (247 tertiary and 49 secondary) students participated. In the activity they were asked to use a tool known as 'Interactive Dichotomous Key (IDK)'. This software assists in plant identification by formulating specific questions: type of reproductive structures, ovary position, connection of floral parts, insertion and shape of the leaves, among others. The authors conclude that the development of multimedia tools such as the IDK can be a simple and effective solution to increase the motivation of students and teachers to study plant-related science. For this learning activity, the researchers used the species *Clematis campaniflora*, *Papaver rhoeas*, and *Ranunculus repens*.

In the selected studies, tertiary school students are an audience with whom almost no plant-based learning activities involving technology are conducted. Due to the scarcity of papers from tertiary level, we are left with speculation as to why this is not occurring. Perhaps the reason for this is that they are already experienced students, and they have a knowledge level where such activities are felt to be inappropriate. Perhaps lecturers find it more appropriate to teach them at an abstract theoretical level. However, researchers should work on innovative learning activity templates to be able to train tertiary students on direct interactions with real natural environments and plant habitats to provide them with new insights through innovative ways of teaching/learning involving plants.

### 3.2.2. Pedagogical Approach

This section classifies the pedagogical approach considered in each learning activity as cited by the authors of the articles reviewed. The most frequently cited pedagogical approaches were *mobile learning* (28.57%) [22,26,30,33], *inquiry-based learning* [27,33,34], and *collaborative learning* [21,27]. In addition, some articles describe learning activities related to *augmented-reality learning* [24], *classroom teaching* [32], *edutainment* [32], *e-learning* [28], *self-learning* [32], and *video learning* [24]. The rest of the authors (35.71%) did not specify the type of pedagogical approach that the learning activity they have performed had [23,25,29,31,34].

With regard to *mobile learning*, Umer, M. et al. [33] conducted an outdoor group activity (4 students) in which 42 secondary school students participated in a unique intervention. Before the outdoor activity, they were asked to do a previous work to understand different concepts related to plants found in the school garden and insert data in an application. Later, they were asked to consult a marker that was placed in the plants with their cell phones that contained information related to the plant and related to the class syllabus. In addition, the application displayed a 3D model of the plant on the marker so that the students could consult information about different parts of the plant. Finally, students filled out a questionnaire and answered questions focused on the mobile learning experience and what they learned from the system. For this learning activity, they used the species *Pinus sylvestris*, *Agaricus bisporus* (fungus), *Arecaceae* (genre), and *Lotus* (genre).

With regard to *inquiry-based learning*, activities are divided into the following phases [35]: 'Orientation', 'Conceptualization', 'Investigation', 'Conclusion' and 'Discussion'. In this review the authors have identified three articles that mention this type of learning [27,33,34], and the authors have been able to verify the phases of each one of them. The 'Orientation' phase is something that all the articles included [27,33,34]; none of them included the 'Questioning' phase; two of them included the 'Hypothesis Generation' phase [27,34]; all of them included the 'Exploration' phase, either planned by the students or the researchers [27,33,34]; two of them included the 'Experimentation' phase, either designed by the students or the researchers [27,34]; two of them included the 'Data interpretation' phase [27,33]; only one of them included the 'Conclusion' phase [27]; two of them included the 'Communication' phase [27,34]; and two of them included the 'Reflection' phase [27,33]. Although many activities try to innovate when it comes to using technology, some are based on traditional teaching methods but incorporate technology. Pinkerton, M. G. et al. [34] conducted an individual *inquiry-based* activity in which 730 primary and secondary school students participated during a unique intervention of 60 min. In the activity, they were asked to conduct a research paper on invasive species and plant biosecurity that they later had to present through self-prepared Powerpoint slides to explain what they had learned. The study was intended to increase the interest and awareness of Florida youth about invasive species so that they would learn concepts related to plant biosecurity and promote early detection of non-invasive species. The activity, divided into three parts (presentation, hands-on activity, and answering student questions), significantly increased the students' understanding of invasive species and the importance of biosecurity.

With regard to *collaborative learning*, Nantawanit, N. et al. [27] carried out an inquiry-based group activity (4–5 students per group) in which 31 secondary school students participated during a semester (6 interventions of 120 min). The activity consisted of several phases: engagement, experimentation, data discussion, active reading, and application, with the motivation of trying to dismantle the belief of many people that animals are more interesting than plants because plants are "passive living beings". This study explored students' perceptions regarding the use of the Fighting Plant Learning Unit (FPLU). In addition, they investigated whether the learning unit influenced the students' interest in the study of plants. For this learning activity, they used the species *Capsicum annuum*.

### 3.2.3. Topic of the Learning Activity

All of the research papers reviewed focused on biology (100%) [21–34]. In addition, one of the articles also focused on the topic of mathematics [25] and another also focused on the topic of computer science [23].

With regard to activities focused on biology, Boudon, F. et al. [23] conducted an activity in which secondary school students participated over 35 weeks. In the activity, they were asked to use L-Py as a training tool in the classroom to construct the 3D plant structure of typical local plants. For this, the students first measured plants in the field, made diagrams, drew the plant architecture, and recorded the spatial distribution of the plants. For this learning activity, they used the species *Euphorbia* (genre).

Very few of these activities have focused on the power of technology to obtain data about the plants themselves. Most of these activities have focused on the subject of biology, and some effort may be needed to get students to understand the ability of technology to obtain data on plants in an objective way. This seems to be related to the school curricula in which the study was conducted, as most of them were primary and secondary schools, as shown in Table 2. At these educational levels, more importance is given to subjects such as biology or knowledge of the natural environment (with a broader and more generalist syllabus) than to others such as computer science (where the syllabus is sometimes reduced to office automation tasks).

### 3.2.4. Purpose of the Learning Activity

The purposes of the learning activities were classified considering the terms as cited by the authors of the articles and extracted from the reviewers in the data extraction process. The main purpose (92.86%) was to *learn about plants, soil, water, and agricultural concepts* [21,22,24–34] (92.86%). Likewise, learning activities (71.43%) were oriented to *train digital competences* [21–24,26,28,30,31,33,34]. In addition, some articles focused on *learning scientific methods* [23,29,31] or *mathematical concepts* [25].

With regard to the *learn about plants, soil, water, and agricultural concepts* purpose, Huang, Y.-M. et al. [26] carried out an outdoor mobile learning activity in which 32 primary school students participated. In the activity they were asked to use personal digital assistants (PDAs) with a Mobile Plant Learning System (MPLS) installed to get information about plants of the environment. With this activity, the researchers investigated the effectiveness of the system to learn about plants.

### 3.2.5. Educational Space Used to Perform the Learning Activity

Figure 2 shows that most learning activities (78.57%) were performed indoors [21–25,28–32,34], whereas 42.86% were performed outdoors [22,26,28–30,33] (implying some kind of field study, i.e., learning activities in real/natural settings). Some activities (28.57%) combined indoor and outdoor spaces [22,28–30]. One learning activity was carried out in a hybrid mode [31].

Zacharia, Z. C. et al. [22] carried out a group activity (4 students per group) in which 48 primary school students participated over six weeks. In the activity they were asked to use smartphones to collect data about outdoor plants. With this, researchers wanted to find out whether using mobile devices to collect plant data improved learning over traditional (notebook and paper) data collection. Those students who used mobile devices were the only ones who were able to appreciate details such as the wind as a pollinating agent thanks to slow-motion recordings.

Plant-learning activities have the advantage that their main object of study, plants, can be easily found anywhere. As such, these types of activities can provide an incentive for teachers to get students out of the classroom and experience different ways to learn about the environment through targeted field studies. However, depending on the type of activity and the level of education, this may mean that the teacher may need help from third parties or parental authorization. Therefore, a good option in primary schools is the use of indoor

plants for students to learn while being in contact with nature. Another option is provided in school gardens and nearby parks.

**Table 2.** Pedagogical approach and learning activity characteristics.

| Reference No. | Education Level | Pedagogical Approach | Topic/Subject Area | Purpose of the Learning Activity | Educational Space |
|---|---|---|---|---|---|
| [26] | Primary school | Mobile learning | Biology | Learning about plants, soil, water, and agricultural concepts<br>Training digital competences | Outdoors |
| [23] | Secondary school | - | Biology<br>Computer science | Training digital competences<br>Learn scientific method | Physical classroom |
| [32] | Primary school | Classroom teaching<br>Edutainment<br>Self-learning | Biology | Learning about plants, soil, water, and agricultural concepts | Physical classroom |
| [24] | Primary school | Augmented-reality learning<br>Video learning (control group) | Biology | Learning about plants, soil, water, and agricultural concepts<br>Training digital competences | Physical classroom |
| [21] | Secondary school | Collaborative learning | Biology | Learning about plants, soil, water, and agricultural concepts<br>Training digital competences | Physical classroom<br>Research laboratory |
| [28] | Secondary school<br>Tertiary school | E-learning | Biology | Learning about plants, soil, water, and agricultural concepts<br>Training digital competences | Physical classroom<br>Outdoors<br>At home |
| [22] | Primary school | Mobile learning | Biology | Learning about plants, soil, water, and agricultural concepts<br>Training digital competences | Physical classroom<br>Outdoors |
| [33] | Secondary school | Mobile learning<br>Inquiry-based learning | Biology | Learning about plants, soil, water, and agricultural concepts<br>Training digital competences | Outdoors |
| [25] | Secondary school | - | Biology<br>Mathematics | Learning about plants, soil, water, and agricultural concepts<br>Learning mathematic concepts | Physical classroom |
| [27] | Secondary school | Collaborative learning<br>Inquiry-based learning | Biology | Learning about plants, soil, water, and agricultural concepts | - |
| [30] | Secondary school<br>Primary school | Mobile learning | Biology | Learning about plants, soil, water, and agricultural concepts<br>Training digital competences | Physical classroom<br>Outdoors |
| [34] | Secondary school<br>Primary school | Inquiry-based learning | Biology | Learning about plants, soil, water, and agricultural concepts<br>Training digital competences | Physical classroom |
| [31] | Secondary school | - | Biology | Learning about plants, soil, water, and agricultural concepts<br>Training digital competences<br><br>Learn scientific method | Physical classroom<br>Research laboratory<br>Remote classroom |
| [29] | Primary school | - | Biology | Learning about plants, soil, water, and agricultural concepts<br><br>Learn scientific method | Physical classroom<br>Outdoors<br>Research laboratory |

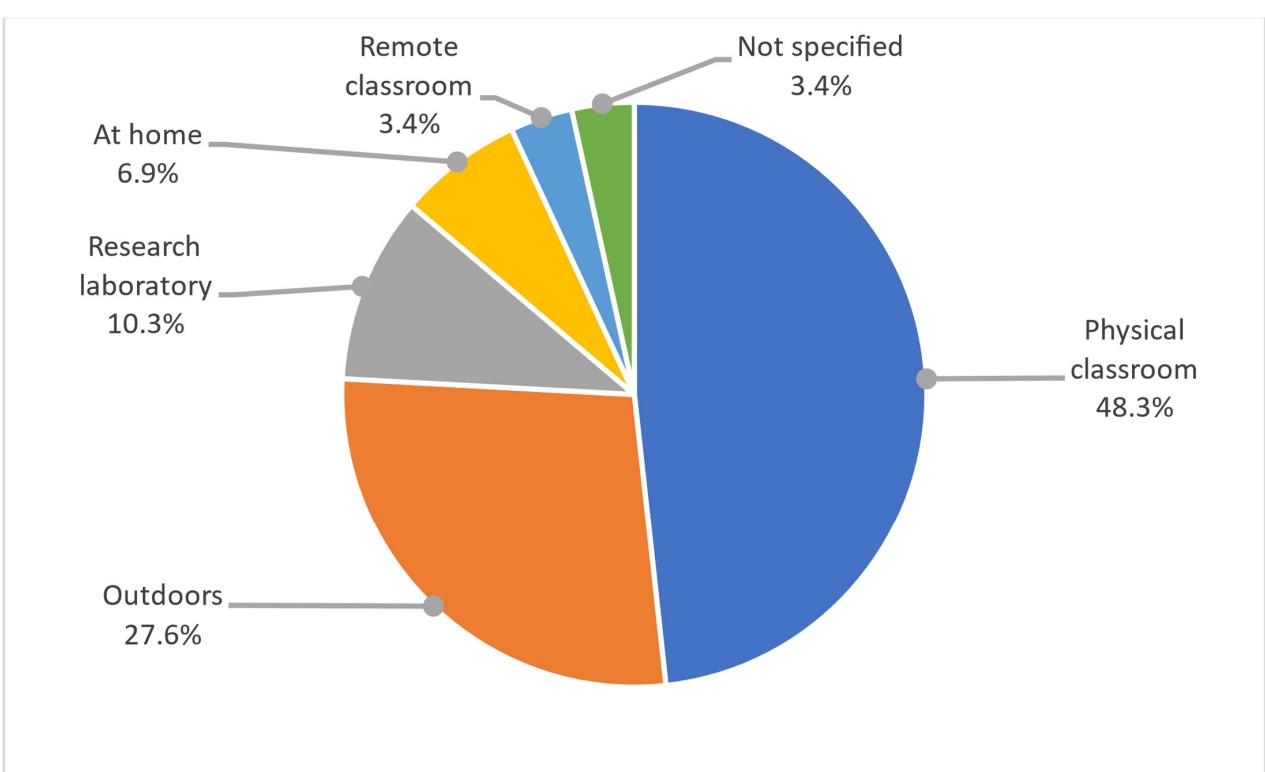

**Figure 2.** Educational space used to perform the learning activity.

### 3.2.6. Organization: Work-in-Groups or Individual

Students worked in groups [21,22,27–29,33] in 6 out of 14 learning activities (42.86%). There were three learning activities where students worked individually [30,32,34]. Two articles described learning activities combining the individual use of the technology within group settings [21,32]. Seven articles (50%) did not specify how students collaborated [23–26,31].

With regard to work-in-groups activities, Pressler, Y. et al. [29] conducted a group activity (2–4 students) in which primary school students were invited to participate in two experiments to explore the effects of biochar on plant growth and soil respiration. Students were invited to observe biochar and different soils through a microscope and different sensors (i.e., $CO_2$ and pH) and discuss properties such as porosity and pH and how these variables can affect the growth of a plant. For this learning activity, they used the species *Vigna radiata*.

With regard to individual activities, Wang, C. [30] conducted an individual outdoor mobile learning activity in which 75 primary and secondary school students participated. In the activity, they were asked to participate in two outdoor mobile learning experiments using an Android application related to plant identification (204 different species) and learning about plants. With this, the researchers wanted to check if the use, learning attitude and interest in the natural sciences were greatly improved after the outdoor activities of plant identification and to explore the extensions that this system can have in different stages of primary school. For this learning activity, they used 204 species, but the authors did not specify what these species were.

Six learning activities (42.86% of the total) involve groupwork. This could be due to several reasons, not necessarily a deliberate pedagogic design. In some instances, researchers may not have had enough technological resources for every student [33], so the students were put in groups. The shortage of technological resources in education centers is a widespread problem for institutions, and, equally, researchers cannot explore the use of technologies to the fullest. Therefore, the development of inexpensive technological systems that could support education should be considered. Based on the results shown in Table 2 and knowing that the studies have been carried out mostly in primary and

secondary schools, it is possible that the students' group work has been influenced by the lack of funding in these centers. It is also worth noting that Nantawanit, N. et al. [27] mention that researchers believe that social interaction among students is one of the main ways to succeed in learning. Therefore, it is possible that this belief may also have played a part in asking students to work in groups.

### 3.2.7. Role of the Stakeholders

This section presents a classification of the roles adopted by students and teachers in each of the learning activities.

One the one hand, the role of the students during learning activities was mainly oriented to perform tasks related to the scientific method (collecting data, analyzing data, or reporting results) (85.71%) [21–23,25–32,34]. Students used technology to create digital content (i.e., using applications, building 3D models, and collecting data, photos or videos) (64.29%) [21–24,26,28,29,31,33] or to interact with the rest of stakeholders (i.e., communication, collaboration, and others) (50%) [21,22,25–27,31,34]. Students identified plants [26,30,31], sometimes using laboratory resources to inspect plants (e.g., magnifying glasses) [22,34] or to take care of the plant [29].

On the other hand, the role of teachers in learning activities was oriented toward assisting students (i.e., suggesting resources based on their knowledge, providing feedback, facilitating materials, or guiding activities) (42.86%) [21,22,26,27,29,31], performing management tasks (e.g., coordinating the learning activity) (28.57%) [21,26,29,34], promoting activities related to scientific method [26,27] or being, primarily, a spotter [24].

Plant-related learning activities are not used solely to learn about plants, but they can also serve as a motive for students to interact with each other (and with others). Thus, the activities can be arranged as a "social activity". However, relating this to the educational levels at which the research has been carried out, it is possible that this occurs because most activities have taken place in primary and secondary school, where this social factor is of great importance, while, on the other hand, such interactions among students has not been reported at the tertiary education level. In addition, the teacher linked to the students did not have a very prominent task within the learning activity. Most of their tasks had to do with supervising the work carried out and providing support where needed. A more active participation of the teacher could lead to higher motivation among the students, which would increase their involvement with the activity. The authors believe that the teacher, too, is a participant in the learning activities. Therefore, if the teacher adopts a passive role within the activities, this could result in a loss of opportunities to investigate new ways of carrying out learning activities or to encourage students' interest.

### 3.2.8. Variables Observed

The following variables were observed in learning activities: (1) academic performance (64.29%) [21,22,24,26–29,31,34]; (2) aspects related to the learning process and its characteristics (learning satisfaction, focus, effectiveness, motivation, or students' perceptions) (28.57%) [29,30,32,34]; (3) aspects related to the technology (acceptance of the tool, engagement of the technology, ease of use, and others) (28.57%) [21,28,30,32]; (4) critical thinking [21,29,31]; and (5) perceived environmental quality [29].

Some authors used pre-existing tests to observe and measure variables: Chang et al. [24] observed *learning motivation* in students using the test developed by Yang et al. [36]; Nantawanit et al. [27] used the Constructivist Learning Environment Survey (CLES) questionnaire developed by Salish [37] to measure learning performance; Pressler et al. [29] used Next Generation Science Standards (NGSS) [38] to measure learning performance. Some articles observed the technologies used: Liu et al. [32] measured technology acceptance [39]; Wang [30] used plant mobile learning applications test to measure location awareness [40].

Some of the evidence indicates that not all research focuses its efforts on transparency, since, although it is possible to deduce what variable is being studied in the research (such as academic performance or technology-related aspects), it is not always clear what type

of protocol or tools the researchers have used to measure these factors. Although there is a possible increase of interest in the use of technology to teach about plants, there are no standardized questionnaires available to evaluate the different characteristics of these methods, other than questionnaires related to the engagement of the students produced by the use of these technologies.

### 3.3. Technolgies Used in Learning Activities with Plants (RQ2)

This section classifies the technologies extracted from the articles (see Table 3). Therefore, technologies used in learning activities with plants are clustered based on whether they are intangible (e.g., software, mobile app, or virtual reality) or tangible (e.g., hardware, smartphones, or sensors).

**Table 3.** Technology used in the learning activities.

| Reference No. | Intangible Technologies | Tangible Technologies |
|---|---|---|
| [26] | PDA app (MPLS: Mobile Plant Learning System) | PDAs |
| [23] | Virtual reality<br>PC app (Blender) | Personal computer |
| [32] | Mixed reality | - |
| [24] | FLARToolKit (AR Library for Flash)<br>Papervision 3D (open-source 3D engine for Flash)<br>Serproxy (middleware)<br>Augmented reality | Arduino<br>Microphone<br>Webcam |
| [21] | Virtual reality<br>Electronic games (Minecraft) | Personal computer (PC)<br>(Alienware Laptop)<br>Oculus Rift |
| [28] | PC app (IDK: Interactive Dichotomous Key)<br>Digitalized gallery (photos and illustrations)<br>Web pages | Personal computer |
| [22] | Smartphone/tablet app<br>(camera and multimedia visualizer apps) | Smartphone<br>Tablet |
| [33] | Augmented reality<br>Smartphone app (MAPILS: Mobile Augmented Reality Plant Inquiry Learning System) | Smartphone |
| [25] | PC app (Photoshop, PaintTool Sai, Geogebra, Microsoft Excel) | Personal computer |
| [27] | Digitalized gallery (photos and videos) | Laboratory tools for chemical reactions |
| [30] | Android app | Smartphone |
| [34] | Powerpoint | Personal computer |
| [31] | App (text processor)<br>Digitalized gallery (images) | - |
| [29] | Digitalized gallery (photos) | Camera<br>$CO_2$ sensor<br>pH sensor |

In this article, we have considered intangible technologies to be those that do not have a physical entity, such as computer programs or technological clusters, as cited by the authors of the selected articles. *Software programs* were the most frequently used technology (64,28%) [22–26,28,30,31,34], followed by *virtual/mixed/augmented reality* (42.86%) [21,23,24,28,32,33], *digitized galleries* (28.57%) [27–29,31], *electronic games* [21], and *web pages* [28]. With regard to tangible technologies, in this article, we have considered tangible technologies to be those that can be manipulated and therefore have a

physical entity. The most frequently used tangible technologies were personal computers (28.57%) [21,23,25,34], followed by devices involving sensors, microcontrollers, or actuators [24,27,29], or smartphones [22,30,33]. In addition, physical devices such as the Oculus Rift [21] and PDAs [26] were featured in learning activities.

The selected articles describe several articles combining both tangible and intangible technologies. With regard to learning activities that used *software*, Zapata-Grajales, F. N. et al. [25] conducted an activity in which secondary school students participated. In the activity, they were asked to perform mathematical modeling of the distribution of leaves on a plant and the growth of the plant. According to the researchers, to carry out this work, the students worked in the same way as foresters do. The activity was divided into two phases. In the first, students constructed geometric models to explain, in mathematical terms, how plants and petals are distributed in plants. In the second, students analyzed plant growth and what mathematical model could model this growth. As a result, they were able to estimate leaf area, simplify difficult-to-obtain quantities (such as area), and thus calculate the plant's ability to receive sunlight. For this learning activity, they used the species *Echeveria elegans*, *Graptopetalum paraguayense*, *Hibiscus rosa sinensis*, and *Lantana camara*.

With regard to learning activities that used *virtual/mixed/augmented reality*, Chang, R.-C. et al. study [24] carried out an augmented reality activity in which 55 primary school students participated during two interventions of 230 and 20 min, respectively. In the activity, students had to use ARFlora, an augmented reality application to learn about six topics: seeding, watering, tropism, day and night changes, photosynthesis, and the relationship between plants and humans. With this experiment, researchers wanted to measure the students' understanding of plant growth and their learning motivation regarding the plant-related curricula. For this learning activity, they used the species *Arachis hypogaea*, *Oxalis corniculata*, and *Mimosa* (genre).

With regard to learning activities that used *digitalized galleries*, Diersen, G. T. [31] carried out a hybrid online activity in which secondary school students participated. In the activity, they were asked to use multiple resources (pollen samples from different plants, digital images of this pollen, and an online digital library) to learn about pollen they had never seen before, using unlabeled samples to avoid associations. Finally, students were required to write a text document that had to be reviewed by other peers, thus fostering an atmosphere of discussion and conclusions among the students. For this learning activity, they used the species *Echinacea angustifolia*.

The universality of certain technologies, such as smartphones, makes them accessible to educational institutions. In cases where the technology is not available in the institutions, the researchers themselves often provide the students with the technologies for the learning activity. The authors suggest that efforts should be made to make informatics a key component of learning activities, potentially by conducting these studies in schools or universities focused on informatics.

*3.4. Plants Explored in Learning Activities Using Technology (RQ3)*

This section aims at classifying the plants used by students during the learning activities. Knowing the plants used and the characteristics that have been explored about them can help to know if there are any selection criteria or educational aspects that help to choose one species or another. Here, papers are classified bearing in mind the following characteristics: species, kind of plantation, features, and measurements gathered from the plant. Some of this information is summarized in Table 4.

**Table 4.** Plants used in the learning activities.

| Reference No. | Species | Kind of Plantation | Features | How Measurements Were Gathered |
|---|---|---|---|---|
| [23] | *Euphorbia* (genre) | Field/Outdoor vegetation | Aspects related to their appearance (height, density, spatial distribution, architecture, width and length) | Manually (measured plants in the field, made diagrams, drew the plant architecture and recorded their spatial distribution) Automatically (built the model in 3D with Blender) |
| [32] | - | Virtual plant (alone, in a pot, in a vegetable patch, or others) | Aspects related to plant requirements (seeding, watering, lighting) Parts or functions of the plant (photosynthesis) | - |
| [24] | *Arachis hypogaea* *Oxalis corniculata* *Mimosa* (genre) | - | Aspects related to their appearance (growth of the plant, tropism, daytime vs. nighttime changes) Aspects related to plant requirements (source of the nutrient, watering, sunlight) | - |
| [21] | *Hyacinthoides* (genre) (virtual) | Virtual plant (alone, in a pot, in a vegetable patch, or others) | Parts related to respiration and photosynthesis and its functions | Manually (observations, impressions, feelings, etc.) (Data obtained through research) |
| [28] | *Clematis campaniflora Brot Papaver rhoeas Ranunculus repens* | Flowerpot or planter (potted plants) | Aspects related to their appearance (multiple parts of plants and flowers) | - |
| [22] | - | School garden | Aspects related to the appearance of a flower (petal, sepal, carpel, and stamen) Parts or functions of the plant (parts mentioned above, related to reproduction and pollination of the plant) | Manually (magnifying glass, notebooks and colored pencils to sketch flowers, notes in a diary) Automatically (through smartphone and tablet camera) |
| [33] | *Pinus sylvestris* *Agaricus bisporus* (fungus) *Arecaceae* (genre) *Lotus* (genre) | School garden | - | - |
| [25] | *Echeveria elegans* *Graptopetalum paraguayense* *Hibiscus rosa sinensis* *Lantana camara* | Field/Outdoor vegetation | Aspects related to their appearance (distribution of leaves or petals, growth) Aspects related to plant requirements (sunlight) | - |
| [27] | *Capsicum annuum* | - | Aspects related to their appearance (plant reaction to parasites and insects) | - |
| [30] | 204 species (not specified) | Field/Outdoor vegetation | Aspects related to their appearance (woody, herbaceous, leaf, shape, flower) | - |
| [34] | - | Greenhouse-raised plants | Aspects related to their appearance (plant health) | Manually (taking notes in a notepad, using magnifying lens) |
| [31] | *Echinacea angustifolia* | Field/Outdoor vegetation | Aspects related to their appearance (pollen) | Manually (slides, word-processing document) |
| [29] | *Vigna radiata* | Flowerpot or planter (potted plants) | Aspects related to their appearance (growth of the plant) Aspects related to plant requirements (pH, $CO_2$, biomass) | Manually (observations impressions, feelings, etc.) Automatically ($CO_2$ and pH sensor) |

### 3.4.1. Species

The results show that there was no preference for specific species. The species chosen are quite disparate and no patterns were identified. It should be noted that the use of physical plants has not always been the case in all studies and, in some cases, students have worked with virtual plants they themselves created. In the experiment conducted

by Southgate, E. et al. [21], students created a virtual *Hyacinthoides* (genre) in Minecraft to explain in detail the parts and functions of the plant.

The authors found a lack of a common pattern in naming species used in learning activities (see Table 4 above). Researchers used different ways to refer to plants, including species name in Latin, genus name, and common name. This inconsistency in naming can lead to confusion and makes analysis of articles difficult. To overcome this, the authors attempted to translate the species name into Latin when there was sufficient evidence to match the name in the article to a specific species.

Most species used have properties such as being resilient or commonly found in certain areas [41–43], making them easy to obtain. However, it is not guaranteed that students will properly care for the plants.

### 3.4.2. Kind of Plantation

The kind of plantations refer to the locations where the plants were situated during the learning activities. Depending on the resources available at educational institutions, these locations can range from potted plants inside the classroom to gardens or plants in the field.

Many learning activities occurred outdoors (6 out of 14) [22,23,25,30,31,33,34]: field vegetation [25,30,31], school garden [22,33], or greenhouse [34]. On the other hand, two articles mention the realization of the learning activity in an indoor space, with potted plants [28,29]. Some researchers encouraged learning activities where the kind of plantation was virtual [21,32]. Four articles (28.57%) did not specify what kind of plantation used in the learning activity [23,24,26,27].

Plant-based learning activities provide opportunities to experiment with alternative teaching methods because plants are widely available. Teaching about natural elements can facilitate non-traditional textbook teaching approaches.

### 3.4.3. Features

Depending on the type of learning activity and its objectives, the features of the plants that students focus on may vary. Therefore, the observed features can be diverse, including the spatial distribution of leaves, nutritional values, and pollen, among others.

As can be seen in Figure 3, students observed different features of the plants during their learning activities. Appearance of plants was the key feature considered when performing learning activities (71.43%) [22–25,27–31,34]. Biological needs was the second most important aspect considered (42.86%) [24,25,29,32]. Lastly, the functions of the plants were the least considered aspect [21,22,32]. The rest of the papers do not specify which parts of the plant the students took into account in the learning activities [26,33].

The focus of most of the articles reviewed on plant-based learning activities was on plant identification, with a significant emphasis on the appearance of the plants. In these activities, students were tasked to examine the details of the plants to classify and identify the species, sometimes relying solely on their own judgment and other times with the support of additional materials [26,28,30,33].

### 3.4.4. Measurements

Table 4 shows how data has been gathered in the articles, whether automatically (such as with the use of sensors) or manually (through annotations, for instance). In six articles, stakeholders manually collected data from the plants (observations, drawings or objective information) (42.86%) [21–23,29,31,34] using digital diaries (text editor) [22,31,34] and paper notebooks [22,23,34]. In three articles, stakeholders automatically collected data [22,23,29] using digital cameras to take photos and videos [22,29] and sensors [29] to measure the pH of the soil and the $CO_2$ of the environment. Out of the total number of fourteen articles, eight of them (57.14%) did not include measurements [24–28,30,32,33].

Pressler, Y. et al. [29] used sensors to measure $CO_2$ and soil pH. Pinkerton, M. G. et al. [34] reports a learning activity in which students observed and annotated subjective measurements

(e.g., taking notes of students' impressions of the plant) about the health of the plant. Plant measurements were not collected in the majority (85.71%) of the articles [21–28,30–33].

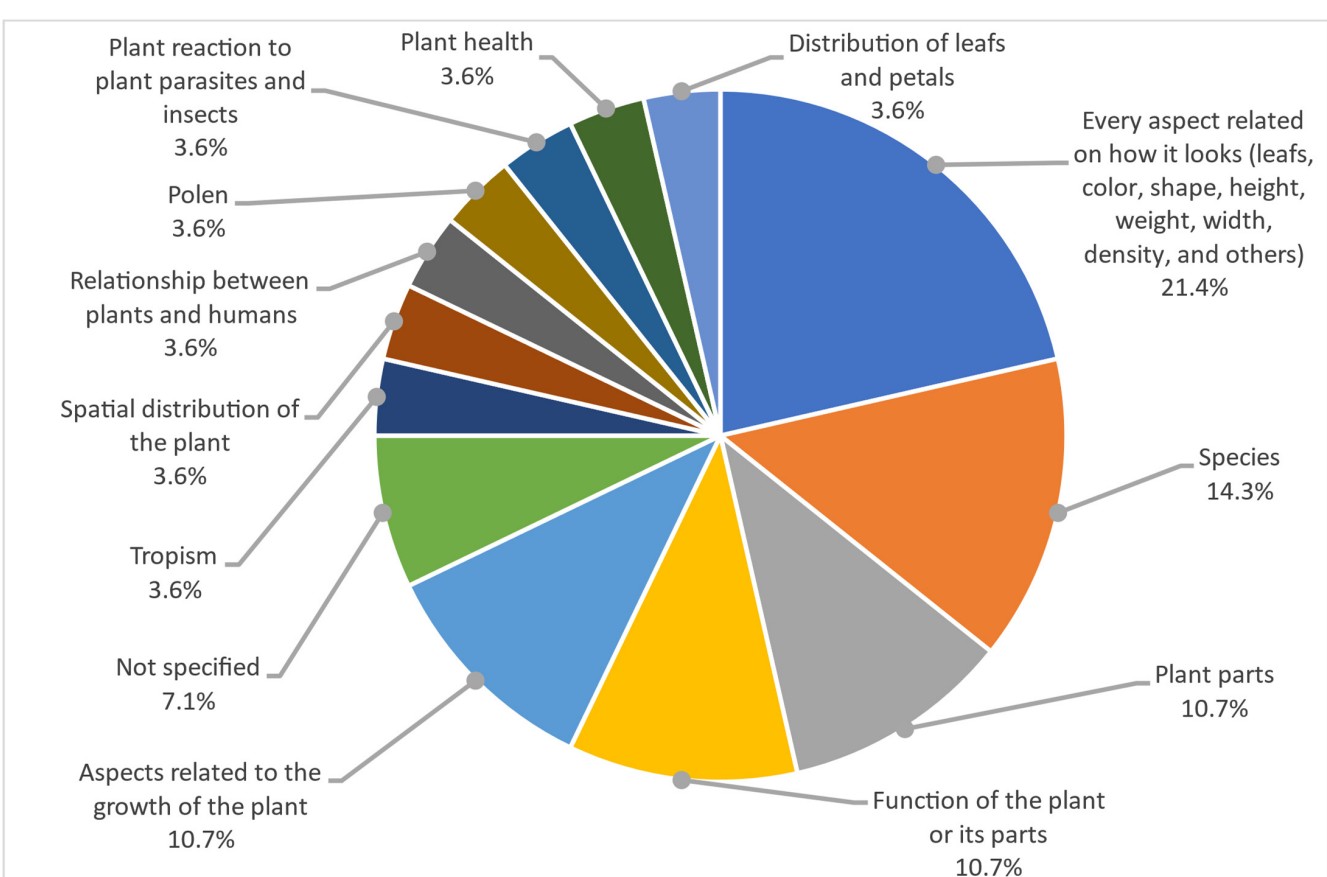

**Figure 3.** Features of the plant species examined during students' learning activities.

## 4. Conclusions

This paper presents a systematic review of the literature on the use of technology in plant-related learning activities. Advances in technology and education must work together for the benefit of all stakeholders, such as students, teachers, and educational institutions.

### 4.1. RQ1 (Pedagogical Approach)

The learning activities studied in the review have primarily taken place in secondary and primary schools, with only one in a tertiary school environment and none in a preschool setting. This may be due to students in tertiary education already having a sufficient level of knowledge and the use of technology being deemed inappropriate or unnecessary for young students. Despite the lack of technological integration in tertiary education, universities have the potential to innovate in education and it is advisable to explore the type of learning activities that could be implemented in these institutions to provide students with technology-based skills.

Apart from that, although the use of plants in learning activities has served multiple purposes (learning about plants, soil, water, and agricultural concepts; training digital competences; learning scientific method; and learning mathematical concepts), none of the reviewed articles have used plants for the purpose of fostering environmental awareness, highlighting a research gap. In the context of the SDGs and the current concern about climate change, the use of plants in learning activities could represent a paradigm shift if the vector of environmental awareness is considered.

In addition, the set-up for learning activities varies for each activity, with some providing a thorough explanation of details such as the number of interventions, duration,

participants, and mode of work, while others offer limited information. It is deemed important for learning activities to be thoroughly detailed in order to facilitate replication by other researchers.

### 4.2. RQ2 (Technology)

Researchers in the field of technology-enhanced plant-related learning activities mainly used applications or virtual reality in their studies. This preference could be due to the convenience of using smartphone applications for outdoor plant identification and the capability of current devices to run augmented reality applications.

The study was conducted using the DigComp framework [44], but the authors note that it does not fully cover the range of technology-related activities that students can perform and does not consider new emerging technologies. Therefore, the authors suggest the need for a new framework that takes into account these advancements. The potential of some emerging technologies is underutilized in education, as indicated in Table 3. Some scenarios may include the use of IoT systems to obtain objective data about the plant and its environment to explain and verify theoretical concepts taught in class; the use of artificial intelligence to predict the behavior or health of the plant; and the simulation of possible scenarios that may positively or negatively affect the plant.

### 4.3. RQ3 (Plants)

The choice of plants for learning activities is based on the researcher's discretion and has primarily been influenced by the selection of local plants or those with high endurance [41–43]. This approach, although based on practical considerations, may not be optimal, as students may not be able to provide the necessary care for high-quality plant growth.

Just as students can interact with each other, it is also possible for students to interact with plants. This interaction usually occurs in a one-way fashion, with students performing actions such as observing or taking notes on the plant. Authors of this review suggest exploring bidirectional communication between students and plants through the use of IoT systems and sensors, which can provide real-time data on the plant's needs and enhance students' learning about plants. This could improve the learning environment by adjusting to both the needs of the plants and the students.

In summary, the use of plants and technology in education is still in its early stages. The studies reviewed in this analysis are not well connected to the field of environmental education or the Sustainable Development Goals (SDGs). The researchers appear to be influenced by the availability of green environments and innovative technological tools, and they come from various geographical backgrounds. Currently, there are no well-established scientific communities in this area. This leaves ample opportunities for future research and pedagogical practise.

**Supplementary Materials:** The following supporting information can be downloaded at: https://www.mdpi.com/article/10.3390/app13063377/s1, Table S1: Number of citations. Table S2: Publication year. Table S3: Role of the students.Role of the students. Table S4: Role of the teachers. Table S5: Role of the researchers. Table S6: Variables observed in the RQ and conclusions. Table S7: Plant/flower features considered. Table S8: Learning activity setup.

**Funding:** This work received partial support from the European Commission through Erasmus+ Strategic Partnerships for higher education through the project TEASPILS (2020-1-ES01-KA203-082258). Likewise, this work has been co-funded by the Madrid Regional Government, through the project e-Madrid-CM (S2018/TCS-4307).

**Institutional Review Board Statement:** Not applicable.

**Informed Consent Statement:** Not applicable.

**Data Availability Statement:** Not applicable.

**Conflicts of Interest:** The authors declare no conflict of interest.

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
