# Peer review of "Learning Activities with Plants and Technology: A Systematic Literature Review"

_applsci, doi:10.3390/app13063377_

Round 1

Reviewer 1 Report

Thank you for submitting your article "Learning activities with plants and technology: A systematic literature review" for review and consideration for publication in the Applied Sciences journal.

This study provides a comprehensive literature review of the use of plants and technology in learning activities using a systematic review. As shown in the number of studies extracted by the authors, this topic is underrepresented in the literature despite the important role of using plants either as 'foreground' or 'background' in the learning process.

The authors did a good job in offering a clear background and rationale for the study, highlighting the importance of caring for nature and the environmental awareness in learning as well as the potential of technology to enhance learning outcomes. The research question is clearly stated, and the scope and methodology of the study are well explained.

The article employs a systematic literature review approach, which is appropriate for the research question. The search strategy is well described and appears to be comprehensive. The article provides clear inclusion and exclusion criteria, and the process for screening and selecting articles is well documented. The quality of the included studies is assessed using appropriate techniques, and I would like to commend the authors for employing the third filtering process to minimize bias. The authors could do better if they discussed the procedures more thoroughly: 

Were the study selection process and any subsequent data extraction or analysis conducted by more than one reviewer to minimize the potential for reviewer bias? If so, what steps were taken to ensure consistency in the review process across reviewers?

Were any efforts made to identify and address potential sources of publication bias in the review, such as by searching for unpublished studies or gray literature? If so, what steps were taken and what was the outcome of these efforts?

"Low quality publications" was part of the exclusion criteria. How did you measure the quality of the publication?

Other than these comments, I believe the paper has merit for publication.

Author Response

We appreciate the reviewer's comments. We have generated a PDF file with responses to all comments made. Please see the attachment.

Reviewer 2 Report

This review paper focuses on the role of technology in learning activities related to plants. In particular, 3 research questions namely, pedagogical approach, technology, and plants have been addressed. The articles have been systematically found and carefully chosen. The paper is well-organized and well-written. Certainly, it would be a valuable reference for the researchers in the this field. Therefore, I would suggest consideration of this work by MDPI Applied Sciences.

-          In my opinion, it is crucial to add one or two graphs to give the readers an overview of the topics being covered in this review article, for instance about the research questions and the methods/criteria. Some of the findings in the form of percentages could also be shown in various types of circle graphs. This way, not only would the paper become more attractive, but the finings would be easier to digest.

-          The Appendix A is pretty long and does not seem to be immediately crucial to comprehend the findings of the paper. It is located before the reference list making it uneasy to refer to a reference while reading the paper. Therefore, I suggest to make it a separate supplementary file.

Minor remarks:

-          Affiliations 1 and 5 are the same; they may be merged. No need to give every author’s email but only for the corresponding author.

-          The title of ref. 29 is written in capital letters.

-          Table 2, the title of the first column Article may be replaced by Reference No. This also applies to other tables.

-          Lines 219 and 244, … a unique…

-          Line 222, was placed in …

-          Linen 253, … an inquiry-based…

-          Line 320, did not specify…

-          Line 327, these variables…

-          Line 354, On the one hand…

-          Line 444, a hybrid online…

-          Regarding article 28 in Table 4, no information is provided; thus, it can be removed from the table.

-          Table 4, article 34, column features: photosynthesis

Author Response

(The authors gave the same response as above.)

Reviewer 3 Report

1. The article should add a discussion section. Because the discussion part is to improve the depth of the article, usually no less than 1000 words.

2. Some important references need to be cited.

Lawal, A.I., Aladejare, A.E., Onifade, M. et al. Predictions of elemental composition of coal and biomass from their proximate analyses using ANFIS, ANN and MLR. Int J Coal Sci Technol 8, 124–140 (2021). https://doi.org/10.1007/s40789-020-00346-9

Gorai, A.K., Raval, S., Patel, A.K. et al. Design and development of a machine vision system using artificial neural network-based algorithm for automated coal characterization. Int J Coal Sci Technol 8, 737–755 (2021). https://doi.org/10.1007/s40789-020-00370-9

Alidokht, M., Yazdani, S., Hadavandi, E. et al. Modeling metallurgical responses of coal Tri-Flo separators by a novel BNN: a “Conscious-Lab” development. Int J Coal Sci Technol 8, 1436–1446 (2021). https://doi.org/10.1007/s40789-021-00423-7

3. There are many small mistakes in the article, including format, punctuation and spelling. Please check carefully.

4. A statement of application prospect needs to be added to the abstract.

Author Response

(The authors gave the same response as above.)

Reviewer 4 Report

Dear Authors

The topic is good for study but the selection of journal is not appropriate as this article is more relevant to the journals of lifelong learning with some revision which are as follows: 

Language is appropriate, very few grammatical mistakes was found that has been corrected. 

Results and discussion should be with more and clear explanation

In figure 1, no of Science direct and Scopus articles are given, the Scopus general mentioned are the excluded one from science direct. 

Self-citation of Bernardo Tabuenca should be reduced to max 2. 

Some corrections and queries has been made on pdf file of manuscript attached. Kindly refer.

Author Response

(The authors gave the same response as above.)

Reviewer 5 Report

 I Accept it in present form. Congratulations to the authors.

Author Response

(The authors gave the same response as above.)

Round 2

Reviewer 3 Report

I think this version can be published at present

Reviewer 4 Report

After reviewing the modification done by the authors, it has been found that all corrections are done.  

Manuscript is accepted in its current form